# Carbon-Based Ternary Nanocomposite: Bullet Type ZnO–SWCNT–CuO for Substantial Solar-Driven Photocatalytic Decomposition of Aqueous Organic Contaminants

**DOI:** 10.3390/molecules27248812

**Published:** 2022-12-12

**Authors:** Santu Shrestha, Kamal Prasad Sapkota, Insup Lee, Md Akherul Islam, Anil Pandey, Narayan Gyawali, Jeasmin Akter, Harshavardhan Mohan, Taeho Shin, Sukmin Jeong, Jae Ryang Hahn

**Affiliations:** 1Department of Chemistry, Research Institute of Physics and Chemistry, Jeonbuk National University, Jeonju 54896, Republic of Korea; 2Department of Chemistry, Tribhuvan University, Amrit Campus, Kathmandu 44618, Nepal; 3Department of Bioactive Material Sciences, Jeonbuk National University, Jeonju 54896, Republic of Korea; 4Department of Physics, Jeonbuk National University, Jeonju 54896, Republic of Korea; 5Textile Engineering, Chemistry and Science, North Carolina State University, 2401 Research Dr., Raleigh, NC 27695, USA

**Keywords:** ternary hetero-composite, photocatalyst, dual Z scheme, heterojunction, organic dyes, carbon-based photocatalyst

## Abstract

A facile two-step synthesis of ternary hetero-composites of ZnO, CuO, and single-walled carbon nanotubes (SWCNTs) was developed through a recrystallization process followed by annealing. A series of nanocomposites were prepared by varying the weight ratio of copper(II) acetate hydrate and zinc(II) acetate dihydrate and keeping the weight ratio of SWCNTs constant. The results revealed the formation of heterojunctions (ZnO–SWCNT–CuO, ZSC) of three crystal structures adjacent to each other, forming a ternary wurtzite-structured nanoparticles along with defects. Enhanced charge separation (electron-hole pairs), reduced band gap, defect-enhanced specific surface area, and promoted oxidation potential were key factors for the enhanced photocatalytic activity of the ternary nanocomposites. OH^•^ radicals were the main active species during dye degradation, and O_2_^−•^ and *h*^+^ were also involved to a lesser extent. A type II heterojunction mechanism approach is proposed based on the charge carrier migration pattern. Among the synthesized nanocomposites, the sample prepared using copper(II) acetate hydrate and zinc(II) acetate dihydrate in a 1: 9 ratio (designated a ZSC3) showed the highest photocatalytic activity. ZSC3 achieved 99.2% photodecomposition of methylene blue in 20 min, 94.1% photodecomposition of Congo red in 60 min, and 99.6% photodecomposition of Rhodamine B in 40 min under simulated sunlight. Additionally, ZSC3 showed excellent reusability and stability, maintaining 96.7% of its activity even after five successive uses. Based on overall results, the ZSC sample was proposed as an excellent candidate for water purification applications.

## 1. Introduction

Organic dyes are widely used in various applications, including textiles, pulp, tanning, cosmetics, food processing, photo-redox catalysis, and dye lasers [1]. Of the dyes used, 50–70% are azo dyes, and 1–20% of all produced dyes reach the environment through wastewater, which can adversely affect the aquatic environment and human health [2]. These dyes are known to be harmful, mutagenic, and carcinogenic if incompletely degraded [1]. Another hazardous amphoteric dye, Rhodamine B (RhB), is a chloride salt and a member of the xanthene family and requires eco-friendly and cost-effective disposal before its release into the environment [3]. Various methods to effectively degrade these dyes have been developed based on adsorption, coagulation, precipitation, filtration, osmosis, or ozonolysis [1,4]. However, these methods are expensive and complex. The use of solar-energy-activated semiconductor photocatalysts to decompose organic dyes to mitigate water pollution via redox reactions has attracted considerable attention because of its advantages, including total mineralization of organic dyes, low operating costs, and high performance in ambient conditions [5]. In addition, the use of semiconductors as green antimicrobial materials is widely emerging [6]. 

Among various semiconductor nanocomposites, ZnO and CuO are currently being extensively studied because of their low cost, non-toxicity, and high thermal and electrical conductivity. ZnO is an *n*-type semiconductor with a bandgap of ~3.44 eV and an exciton energy of 60 meV, widely applied in optoelectronic devices such as laser diodes, photodetectors, and photosensors [6,7,8]. However, due to the large band gap, ZnO can only be stimulated by ultraviolet (≈5% under sunlight), which limits its activity under visible light. In order to overcome these shortcomings, various efforts have been made to modify ZnO with other substances (e.g., metals, nonmetals, semiconductors, or carbon-based materials such as graphene or carbon nanotubes). The introduction of these materials into ZnO enables control of the bandgap and improves the separation of charged particles. For example, when ZnO is combined with other visible light active semiconductors such as CuO, the band gap is reduced, and the photoresponse range is extended through heterojunction formation within the composite. CuO is a *p*-type photoactive material with a bandgap of 2.1 eV [9,10]. Accordingly, the ZnO/CuO combination improved the catalytic activity due to the improved transport of charge carriers through heterojunction [6]. Although the heterojunction of ZnO and CuO shows improved catalytic activity over the individual components, the photocatalytic performance has not yet reached the mark. The photocatalytic performance is still insufficient due to the high electron–hole (*e*–*h*) recombination of CuO in the ZnO-CuO heterojunction and the band gap mismatch between CuO and ZnO. Therefore, ZnO-CuO must be combined with a highly conductive material to increase the photocatalytic activity under sunlight. Finding a suitable photocatalytic compound that works well in sunlight seems to be still a challenge.

Carbon-based materials, as charge carriers with high conductivity, can be chosen as a solution to this problem. In particular, single-walled carbon nanotubes (SWCNTs) have high conductivity, photoactivity over a wide spectral range, and a relatively larger specific surface area [9]. They increase the specific surface area of heteronanocomposites, introduce surface defects, and act as an electron shuttle to facilitate charge transfer and lower the *e*–*h* recombination rate. However, excessive amounts of SWCNTs may be undesirable because they will reduce the exposure of the semiconductor surface to incident light. Thus, SWCNTs can be utilized as bridges to cure poor charge carrier migration and photon absorption. Enhanced charge carrier migration through SWCNTs can reduce *e*–*h* recombination in CuO. SWCNTs can also enhance *e*–*h* separation by photosensitization. Although several studies have been reported in which combinations of SWCNTs with ZnO or CuO are applied to the decomposition of pollutants, the photocatalytic efficiency of composites is still not satisfactory due to inadequate charge separations [5,9,10,11]. 

Based on the above hypothesis, we attempted to combine SWCNTs with ZnO and CuO, and finally successfully synthesized ZnO-SWCNT-CuO (ZSC) nanocomposite through a two-step facile method. The synthesized composites showed excellent photocatalytic activity against dye-containing wastewater in simulated solar irradiation. Here, CuO plays the role of visible light absorption, band gap reduction, and photo-corrosion reduction, and SWCNTs act as charge carrier media to retard the electron-hole recombination rate. The physicochemical properties of the synthesized ZSC nanocomposites were systematically investigated by various spectroscopic, microscopic, and electrochemical techniques. The best composition nanocomposite showed superior activity for methylene blue (MB) degradation when compared to mono or dual component materials. In addition, the catalytic activity for photodegradation of Congo red (CR) and RhB dyes also showed remarkable efficiency. The enhanced photoactivity of ZSCs is attributed to the efficient separation of photoexcited charge carriers, reduced charge carrier recombination rate, well-developed surface defects, extended specific surface area, and promoted redox potential. Moreover, the photocatalytic activity remained unchanged even after 5 consecutive runs. Based on various characterizations and analyses, we propose a type II heterojunction mechanism. 

## 2. Results and Discussion

### 2.1. Crystallographic and Spectroscopic Analysis

The ZSC heteronanocomposites ZSC1, ZSC2, and ZSC3 were analyzed by X-ray diffraction (XRD) to characterize their crystallinity and crystallite size. Copper(II) acetate hydrate (CAH) and zinc(II) acetate dihydrate (ZAD) were combined in ratios of 1:3, 1:6, 1:9, and 1:12 by mass for the synthesis of ZSC1, ZSC2, ZSC3, and ZSC4, respectively. The diffraction patterns of pristine CuO and pristine ZnO were also collected as reference patterns to aid in the characterization of the crystallinity of the monoclinic CuO and wurtzite ZnO in the ZSC heteronanocomposites. In Figure 1a, the 2*θ* peaks at 43.02°, 50.04°, and 73.86° in the diffraction peak of pristine CuO correspond to the (111), ($\bar{2}$02), and (222) facets of monoclinic CuO, respectively [10]. In the XRD plot of the pristine ZnO, several diffraction patterns at 2*θ* values of 31.67°, 34.31°, 36.20°, 47.48°, 56.47°, 62.75°, 66.3°, 67.9°, 68.9°, and 76.82°, which are assigned to the (100), (002), (101), (102), (110), (103), (200), (112), (201), and (202) crystalline facets of wurtzite-structured ZnO (JCPDS No. 36–1451) [6], were observed. The diffraction patterns for heterocomposites ZSC1, ZSC2, and ZSC3 in Figure 1a show all the peaks corresponding to monoclinic CuO and hexagonal wurtzite ZnO, revealing that the crystallinity of both oxides was preserved in the as-synthesized nanocomposites. Figure 1b shows a magnified XRD crystalline peak of carbon nanotubes in the heterocomposites, confirming the presence of SWCNTs in the as-prepared samples [12].

The average crystallite size (*D*) of ZSC samples were calculated from XRD spectra by using Scherrer’s Equation (1):(1)$$D=\frac{k\lambda }{\beta \cos\theta }$$
where *k* is the Scherrer constant (*k* ≈ 0.9), *λ* is the wavelength of the X-rays (*λ* = 0.154 nm for Cu K*α* radiation), *β* is the full-width at half-maximum of the peak in radians, and *θ* denotes the diffraction angle in radians [10]. The average crystallite sizes calculated for ZSC1, ZSC2, and ZSC3 using Gaussian fitting with the most intense peak (101) plane, are 19.7 nm, 24.5 nm, and 27.6 nm, respectively. The increasing concentration of ZnO (ZSC1 ˂ ZSC2 ˂ ZSC3) in ZSC samples may be responsible for the increasing order in crystallite size.

Fourier transform infrared (FTIR) spectroscopy was employed to probe the chemical bonding and determine the functional groups present in the samples. Appendix A shows the FTIR spectra of the pristine wurtzite ZnO, pristine monoclinic CuO, and the ZSC heteronanocomposites in the wavenumber region 400–4000 cm^−1^. The broad band at 3450 cm^−1^ and the peak at 1630 cm^−1^, which are ascribed to the stretching and bending vibrations of O–H groups, respectively, indicate the formation of hydrogen bonds [6]. The peak at 2348 cm^−1^ is assigned to the strong stretching of atmospheric CO_2_. The characteristic stretching band for the monoclinic phase of CuO was observed at 664 cm^−1^ in the spectra of all the heteronanocomposites and was similar to the corresponding band in the curve of pristine CuO [13]. The peak for Zn–O stretching was observed from 401 to 508 cm^−1^ in the curve of pristine ZnO [13]. The broad band observed in this region for ZSC samples is the result of a distortion of the Zn–O bonds due to the presence of CuO. ZSC1 shows greater band broadening than the other nanocomposites because of its greater CuO content [14].

### 2.2. Morphological Characterizations

The morphological characteristics of the as-synthesized ZSC heteronanocomposites were evaluated by field-emission scanning electron microscopy (FE-SEM) (Figure 2). Figure 2a displays an FE-SEM micrograph of pristine ZnO with needle-like structure, and the inset shows the monoclinic-like structure of CuO. Figure 2b,c show varied structures of nanocomposites ZSC1 and ZSC2, respectively, whereas Figure 2d shows a defective surface structure of nanocomposite ZSC3. ZSC3 has kinks on the surface, which was further confirmed by high-resolution transmission electron microscopy (HR-TEM) [15]. These kinks increase the surface area, which promotes charge polarization, resulting in enhanced photocatalytic activity [16]. The FE-SEM image of ZSC1 shows that a moderately developed wurtzite-like structure of nanocomposite aggregated to develop into an oyster mushroom-like structure. Figure 2e shows a micrograph of the ZSC3 heteronanocomposite, and Figure 2f displays the size distribution of the ZSC3 heteronanocomposite for 100 particles, as computed using the ImageJ software. The average diameter of the ZSC3 nanocomposite particles, as seen in the histogram (Figure 2f), is 72.7 nm.

Energy-dispersive X-ray spectroscopy (EDS) was performed to explore the chemical elements and purity of the as-prepared nanocomposite materials (Appendix A). The figure shows that Zn occupies the surface to a greater extent than the other elements (O, C, and Cu) present in the heteronanocomposites. The inset in Appendix A displays the distribution of the elements available in ZSC3.

The morphology of the as-prepared ZSC nanocomposites was further examined by HR-TEM to provide evidence of the presence of ZnO–SWCNT–CuO heterojunction (Figure 3). The white mark in Figure 3b, which is a magnified micrograph of Figure 3a, shows the presence of SWCNTs in the nanocomposite. The HR-TEM images in Figure 3c,d reveal discontinuous lattice fringes of ZnO, CuO, and carbon nanotubes present in the composite. Among the lattice stripes, the interplanar *d*-spacing of 0.25 nm matches the (11$\bar{1}$) plane of monoclinic CuO [17] and that of 0.285 nm corresponds to the lattice fringe of the (100) plane of ZnO [15]. Interplanar spacings of 0.54 nm, which corresponds to the (01$\bar{1}$0) facet of the hexagonal structure of ZnO [18], and 0.313 nm, which corresponds to the lattice spacing of carbon nanotubes [19], were also observed. The TEM micrograph in Figure 3d supports the formation of a heterojunction of ZnO–SWCNT–CuO. Figure 3e,f show the presence of SWCNTs with a hexagonal wurtzite structure and substantial defects, respectively.

The chemical elements and electronic structures of the samples were further examined by X-ray photoelectron spectroscopy (XPS) (Figure 4a–f). Figure 4a shows the survey scans of pristine samples and the synthesized heteronanocomposites. The spectra of all the nanocomposites show the presence of Zn and Cu in the (+II) state, along with C-1s peaks originating from the SWCNTs. Figure 4b shows Zn-2p XPS plots of pristine ZnO and the heteronanocomposites. The 2p_3/2_ and 2p_1/2_ peaks were observed at 1021 and 1044 eV, respectively. The spin-orbit splitting value of 23 eV strongly supports the presence of Zn^2+^ in the nanocomposites [6]. The peaks in the nanocomposites’ spectra show a shift toward lower binding energies compared with the peaks in the scan of pristine ZnO. The shift in energy for the nanocomposites is attributable to the combination of ZnO, SWCNTs, and CuO.

Figure 4c displays the Cu-2p XPS curves of all the heterocomposites and the pristine CuO. The figure shows Cu-2p_3/2_ peaks at a binding energy of 932.63 eV for the pristine CuO [20]. The Cu-2p_3/2_ binding energies for the composites are similar to those previously reported for other CuO-containing nanocomposites [6], with shake-up peaks characteristic of CuO [6,10]. Another spin-orbit coupling is observed at ~952.48 eV for Cu-2p_1/2_, which is also red-shifted to a lower binding state in the spectra of the heterocomposites [10]. The existence of Cu^2+^ ions in pristine CuO is also supported by the monoclinic structure observed in the FE-SEM image (inset of Figure 2a). The XPS results strongly support the presence of Cu^2+^ ions in the as-synthesized heterocomposites. The spin-orbit separation is 19.48 eV, consistent with that of the literature [6]. The shift in binding state for the Cu-2p_3/2_ and Cu-2p_1/2_ peaks in the spectra of the heterocomposites might be a result of nanoparticle structural defects resulting from the ZnO, CuO, and SWCNTs. The largest shift was observed for ZSC2.

Figure 4d–f show the O-1s spectra for the ZSC1, ZSC2, and ZSC3 heterocomposites, respectively. The peaks were observed at approximately 530, 531, and 532 eV in the scans of all the heteronanocomposites, revealing the presence of O^2−^, oxygen vacancies, and oxygen due to chemisorbed species such as –OH^−^, H_2_O, or O_2_^2−^. The intensities of the peaks of oxygen vacancies and chemisorbed O are much lower than that for the peaks of O^2−^ in the spectra of the ZSC1 and ZSC3 nanocomposites, whereas the spectrum of ZSC2 shows intense peaks for oxygen vacancies and chemisorbed O. The C-1s core-level spectra of ZSC1, ZSC2, and ZSC3 are shown in Appendix A. These spectra show the peaks related to C=C (sp^2^), C–C (sp^3^), and –O–C=O at approximately 283, 285, and 289 eV, respectively, which matches the previously reported values [5].

### 2.3. Thermal Properties of Nanocomposites

The thermal strength and stability of the as-prepared heteronanocomposites were analyzed via thermogravimetry/differential scanning calorimetry (TGA/DSC) measurements. Both the nanocomposite and pristine samples were annealed from 27 to 850 °C at a ramp rate of 10 °C min^−1^. The heating atmosphere was O_2_ for SWCNTs and N_2_ for all other samples, including pristine samples of ZnO and CuO. The SWCNTs were annealed in O_2_ atmosphere to determine its purity, whereas N_2_ atmosphere for other samples was used to avoid oxidation. The results are presented in Figure 5 in the form of graphical plots. In Figure 5a, a slight mass loss of pristine CuO is recorded at 116–228 °C, which is due to the evaporation of physisorbed water on the sample surface [21]. A moderate weight loss was observed from 391 to 566 °C, which corresponds to the decomposition of unreacted acetates remaining in the sample [20]. A similar rate of decomposition was observed from 566 to 850 °C, resulting in 96.3% of residual CuO [21]. Pristine ZnO exhibited the highest thermal stability, with a 99.4% residual amount. A slight mass loss of 0.26% from pristine ZnO was detected from 283.7 to 401.0 °C because of the loss of physisorbed water and a loss of 0.4% from 401 to 850 °C was due to the dissociation of unconsumed zinc acetate remaining in the sample [5]. Among the three heteronanocomposites, ZSC3 exhibited the greatest stability (96.0%) and ZSC1 exhibited the lowest stability (92.6%), which is attributable to the increasing ratio of ZnO and decreasing ratio of the SWCNTs from ZSC1 to ZSC3. The minimal losses observed for the ZSC samples are the result of the volatilization of water adsorbed onto their surface, the removal of hydroxyl groups, and the dissociation of the SWCNTs [22]. Figure 5b supports the preceding statement. It shows the extensive decomposition of SWCNTs with increasing temperature. The residual amount of SWCNTs was only ~3% when the SWCNTs were heated to 850 °C, with a particularly large mass loss in the temperature range 519–584 °C and a sharp loss at 542 °C. The high stability of the heteronanocomposites is attributable to their chemical and physical stability, the nearly complete decomposition of the precursor acetates during their synthesis, and the almost complete removal of the water remnants as a result of the applied heat treatment [22]. The thermograms in Figure 5c,d show the endothermic nature of the as-prepared samples.

### 2.4. Photoluminescence and Optical Characterization

Bandgap control and crystal defects are key factors for enhancing the photocatalytic activity of semiconductors [16]. Crystal defects in a favorable concentration range enhance the activity of photocatalysts. Photoluminescence can determine radiative as well as nonradiative transitions qualitatively and quantitatively [14]. Intrinsic emission lines arise because of free excitons, whereas extrinsic emission lines are related to donor, acceptor, and defect states. Defect states can occur because of the synthesis method, the presence of impurities, and/or the introduction of a dopant [23]. 

The defect states of the synthesized samples were analyzed using PL spectroscopy. Figure 6 displays the deconvoluted PL spectra obtained for the heteronanocomposites and pristine oxides of ZnO and CuO. Broad peaks in the regions ~387 to 422 nm and from 455 to 535 nm were detected in the PL scan of all the samples, which are attributable to the recombination of photogenerated carriers, while the peaks that appeared at ~413, 482.48, and 527.9 nm in the 455–535 nm region are likely attributable to an indirect emission caused by defects related to interstitial Zn [6]. The peaks around ~482 nm are attributed to deep-level emission [16]. The peak at ~413 nm in the scan of pristine ZnO and the broad peak in the spectra of the nanocomposites and pristine CuO are likely attributable to band edge emissions [6]. The peak at ~527.9 nm in the curve of ZSC3 is likely attributable to the presence of singly ionized oxygen vacancies [16]. The emission is observed as a result of radiative recombination between holes and electrons occupying the oxygen vacancies generated by photons. The PL spectra of the heteronanocomposites containing ZnO, SWCNTs, and CuO show a substantial reduction in PL intensity compared with the spectrum of pristine ZnO, indicating a slowed *e*–*h* recombination process, and the spectrum of the ZSC3 sample shows the lowest PL intensity among all the spectra. The bandgap energies determined from the PL spectra for all the synthesized samples were found to be in good agreement with those determined from Tauc plots (Figure 7), confirming that ZSC3 has a minimum bandgap of 3.052 eV, corresponding to the band edge emission wavelength of 405.76 nm. The combined PL spectra for all the samples are provided in Appendix A.

The *e*–*h* pair separation efficiency of the ZSC3 was further investigated by electrical impedance spectroscopy (EIS) and cyclic voltammetry (CV) measurements (Figure 8). The EIS plot of ZSC3 and pristine ZnO is shown in Figure 8a. The Nyquist plot of ZSC3 has a much smaller arc radius than pristine ZnO indicating the substantial reduction of interfacial charge-transfer resistance. The reduction of interfacial charge-transfer resistance helps to enhance photocatalytic activity by creating a fast electron-hole separation [24]. CV was performed from −1.2 V to 0 V range to investigate the electrochemical properties of pristine ZnO and ZSC3. CV curves for ZSC3 and pristine ZnO are shown in Figure 8b. The higher oxidation potential was observed for ZSC3 than in the pristine ZnO sample, indicating the enhanced electrooxidation capacity of ZSC3. CV curves were recorded for ZSC3 at different scan rates. The steady increase of current density at high scan rates indicates the high stability of the as-synthesized ZSC3 nanocomposite.

The optical features of all the synthesized samples were studied by UV–vis spectrophotometric analysis (Figure 7). The absorption peak in the plot of the as-prepared pristine ZnO appears at 361 nm, and absorption peaks in the plots of the ZSC samples are red-shifted compared with that in the spectrum of the pristine ZnO [5]. The red shift of the peak is attributable to the modified electronic structure, bandgap lowering, and improved crystallinity of the ZSC samples as a result of the incorporation of CuO and SWCNTs for heterojunction formation in nanocomposites [6]. The absorptions were detected at 264 and 260.8 nm in the curve of the SWCNTs and pristine CuO, respectively [5,20].

The optical bandgap (*E*_g_) of all the samples, including pristine samples, was analyzed using Tauc plots. Figure 7 shows the direct-bandgap Tauc plots for all the samples, including pristine ZnO. Among the ZSC nanocomposites, ZSC3 exhibited the smallest bandgap of 3.06 eV. The bandgap in a heteronanocomposite changes as a result of structural reconstruction, modification of the electronic structure, or alteration of the particle size [16]. The Fermi level in the bandgap of a component is altered when it is combined with a guest material. A *p*-type material replaces the VB, and an *n*-type material replaces the CB, resulting in bandgap lowering [9]. Both direct and indirect bandgap methods were used to characterize the bandgaps of ZSC1, ZSC2, and ZSC3. The Tauc plots obtained by the indirect-bandgap method (Appendix A) were used to evaluate the nature of the bandgap in the prepared ZSC samples. The results confirmed that the ZSC samples have a direct bandgap. Figure 7d shows the Tauc plot for pristine CuO, as obtained using the indirect-bandgap method.

### 2.5. Photocatalytic Activity

All of the synthesized nanocomposites were studied for their photocatalytic activity toward MB degradation with respect to time (Figure 9). In each experiment, 100 mL of a 25 ppm MB solution in distilled water was used. A 50 mg dose of each of the ZSC samples was added to the MB solution and dispersed by sonication, followed by magnetic stirring. Adsorption–desorption equilibrium was achieved by placing the reaction mixture in the dark for 1 h. The mixture was then placed in a homemade simulated sunlight device equipped with a Xe lamp (OSRAM XBO R, 300 W/60C, 0.11 W/cm^2^) that reproduces the full solar spectrum. The distance from the light bulb to the reaction mixture in the instrument was 15 cm. The experiments were conducted at room temperature (25 °C). Homogenization of the reaction mixture was achieved using a magnetic stirrer at an agitation rate of 80 rpm. Samples (3 mL) were collected from the mixture at 5 min intervals to determine the photocatalytic degradation rate of MB. However, in the case of ZSC2 and ZSC3, samples were collected in 5 min intervals initially until 10 min and were then collected at shorter time intervals of 3 and 2 min to monitor the degradation process precisely (Figure 9d–e). 

The photodegradation of MB can be enhanced by the generation of singlet oxygen ^1^O_2_ in the presence of visible light by self-photosensitization [25]. Triplet excited state MB (3MB*) on reaction with dissolved oxygen produces ^1^O_2_ by energy transference. The resulting singlet oxygen ^1^O_2_ reacts with the substrate through Type II photosensitization. Triplet excited state MB (3MB*) also reacts with the substrate via Type II photosensitization. In order to confirm this effect, the photodegradation of MB was observed in the absence of a photocatalyst. Appendix A shows the photodegradation of MB is 8.9%, ©©h is attributed to the self-photosensitization of MB by absorption of visible light [25].

Among all the nanocomposites, ZSC3 exhibited the best activity for the complete decomposition of MB. Figure 9a–e show the absorption spectra collected during MB photodegradation in the presence of ZSC samples and pristine samples. The absorption spectrum of MB shows a strong peak at ~661 nm due to monomeric MB, with a lower-intensity shoulder peak at 611 nm due to the dimeric MB [16,21]. Additionally, two absorption bands are recorded at ~248 and ~292 nm in the UV range. These bands are due to the substituted benzene rings of MB [26]. In Figure 9e, the time-function spectra of ZSC3 for the photocatalytic degradation of MB show that all the absorption bands disappeared completely under exposure for 20 min in the solar light simulator, indicating complete degradation of MB. A kinetic study of the photocatalytic decomposition of MB by the ZSC nanocomposites affirmed that the reaction followed pseudo-first-order kinetics. MB degradation kinetics was also studied for pure TiO_2_ (Degussa p25) under the same reaction conditions for comparison. Appendix A shows a kinetic plot for MB photodegradation by pure TiO_2_ (Degussa p25). Table 1 shows the calculated *k* values for various samples under simulated sunlight. The *k* values for MB degradation by ZSC1, ZSC2, and ZSC3 heteronanocomposites were 0.063, 0.112, and 0.198 min^−1^, respectively, which are 2331%, 4163%, and 7335% greater than *k* value of the blank experiment (0.0027 min^−1^). In addition, the *k* value of ZSC3 is 2.01 times (201%) greater when compared to the activity of commercial nanoparticles (Degussa p25). ZSC3 exhibited excellent MB photodegradation performance of 99.2% with outstanding stability, demonstrating 96.7% degradation even at the fifth cycle. The spectrometric photocatalytic activity studied for each specimen, a time-function graph of MB concentration, a kinetic plot using the pseudo-first-order rate equation, and the recycle stability of ZSC3 are presented in Figure 9.

In order to verify the reusability of the photocatalyst and the potential leaching of zinc oxide and copper oxide, an XRD pattern of the used photocatalyst (after four runs) was investigated. Comparative XRD plots of the fresh and used photocatalysts are shown in Appendix A. The XRD plot obtained from the used photocatalyst shows no significant change in crystalline structure. These results also show that there are no erroneous results due to handling and slow corrosion [27].

Figure 10a shows UV–Vis plots recording during the decomposition of CR dye in the presence of the ZSC3 nanocomposite. The spectrum at zero time of irradiation shows three peaks at 497, 346, and 235 nm. The peak at 497 nm is assigned to the π → π* transition of the azo group, and the other two peaks correspond to the π → π* transitions of the –NH group of the naphthalene ring [28]. The spectra show a subsequent decrease in the intensity of the peak at 497 nm with increasing irradiation time, along with a green shift, resulting in complete decolorization of the dye and confirming the breaking of the azo bonds by hydroxyl radicals. The peaks at 346 and 235 nm observed at zero time were shifted to 375 and 264 nm, respectively, after 60 min of irradiation, which is a result of the formation of aromatic intermediates during the decomposition of the dye. The bands at 346 and 235 nm, which were intensified upon initial irradiation, are attributable to a naphthalene-substituted derivative and a disubstituted benzene derivative, respectively [28].

Figure 10b shows the photodegradation spectra of RhB by ZSC3 as a function of time. The absorption spectrum at zero time shows a typical band at 552 nm because tetraethylated rhodamine exhibits a characteristic S_0_–S_1_ transition in its xanthene ring, as previously reported [3]. The absorption peak at 519 nm is attributable to the fully dehydrated RhB molecule. Complete decomposition was achieved at 40 min, as indicated by a visual color change of the dye solution from pink to greenish-yellow, followed by the observation of a colorless solution, which was confirmed by spectrometric determination. A blue shift from 552 to 520 nm was observed at 30 min, with complete disappearance of the absorption peak at 40 min, possibly because of structural modifications and the disappearance of chromophore groups [3]. A delay in the recombination of separated *e*–*h* pairs and enhanced availability of charge carriers for the oxidation of RhB might be responsible for the effective decomposition of the RhB. The *k* values of CR and RhB degradation by ZSC3 are 0.046 and 0.137 min^−1^, respectively (Table 1). Kinetic plots for CR and RhB are presented in Appendix A. Appendix A shows the UV-vis photolysis spectra of CR and RhB.

The photocatalytic performance of ZSC3 was compared with those of similar nanocomposites in Table 2. In comparison, the ZSC3 nanocomposite outperforms other composites in terms of time and dye-to-catalyst dose ratio. In addition, our as-synthesized ZSC samples showed substantial photoactivity in simulated sunlight instead of UV as reported in most of the other studies. The ZSC3 composite shows performance in only 20 min for 25 ppm MB degradation, 60 min for 25 ppm CR, and 40 min for 25 ppm RhB in the presence of 50 mg of the composite.

### 2.6. Determination of Point of Zero Charge and pH

The drift method was employed to determine the pH_PZC_ for the ZSC3, heteronanocomposite with optimized optical activity (Appendix A). The pH_PZC_ value determined for ZSC3 is 8.93, revealing that the heteronanocomposite is negatively charged at pH levels greater than 8.93 and positively charged at lower pH values [29]. Since the MB dye used in our study is positively charged, the pH_PZC_ of ZSC3 indicates that adsorption is more favorable at higher pH levels. Additionally, the presence of OH^−^ ions at high pH values boosts the reaction by facilitating the generation of OH^•^ radicals.

### 2.7. Scavenging Experiments

The dyes were photocatalytically degraded as a result of their interaction with the active charge carriers produced under exposure to light. Highly influential radicals such as OH^•^, O_2_^−•^, and *h*^+^ are known to be involved in the reaction [6]. In order to identify the most influential reactive species, the photocatalytic reaction was performed with the optimized heteronanocomposite ZSC3 in the presence of isopropanol, potassium iodide, and 1,4-benzoquinone, each separately. The experimental parameters were the same as those employed in the photocatalytic decomposition of MB except for the addition of 3 mL of isopropanol (to trap OH^•^), 5 mg of potassium iodide (to scavenge *h*^+^), or 5 mg of 1,4-benzoquinone (to scavenge O_2_⁻^•^). Appendix A shows the results as the ratio of the decrease in the concentration of the dye as a function of time. The results reveal that OH^•^ was the most influential reactive species, whereas O_2_⁻^•^ was the least influential species. These results agree with the experimental pH_PZC_ data. During the experiment, 79.6% of the MB dye remained undecomposed under exposure to radiation for 20 min in the presence of IPA, whereas only 4.0% and 54.4% of the MB was undegraded in the presence of 1,4-benzoquinone and potassium iodide, respectively. The inhibition rate (%) exhibited by scavengers: IPA, KI, and BQ is 95%, 86%, and 29%, respectively (Appendix A). Table 3 shows the rate constant and inhibition rate (%) of MB photodegradation reaction mediated by different scavengers (obtained from Appendix A).

### 2.8. LC-MS Analysis

LC-MS experiments were conducted to investigate the degradation of the dye during photocatalysis. MB was chosen as a representative dye. Appendix A shows the mass spectra recorded for MB before and after photocatalysis under simulated light for 20 min. Appendix A shows a strong peak at *m*/*z* = 284, consistent with the presence of MB in the solution. The peak of MB (*m*/*z* = 284) completely disappeared, and several peaks associated with intermediates were newly observed in Appendix A. The structure of MB was broken into numerous compounds with low molecular weight. The degradation process is proposed in Appendix A. The results confirm the effectiveness of the ZSC heteronanocomposite for dye degradation. Regarding CR and RhB, several references reported that the intermediates states of CR and RhB are less toxic than the elementary dye molecules [30,31,32].

### 2.9. Photocatalytic Mechanism

In order to elucidate the mechanism of the photocatalytic reaction, the *E*_g_ values of the heteronanocomposite were evaluated using Tauc plots. The UPS spectra were obtained to determine the valence band maxima of ZnO and CuO (Appendix A). Along with the band gap (*E*_g_), UPS spectra can be used to determine the conduction band minimum [33]. The empirical formula in Equation (2) is employed to determine the CB minimum (*E*_CB_).
(2)$$E_{\mathrm{VB}}=E_{\mathrm{CB}}+E_{g}$$

The obtained valence band maxima for ZnO and CuO from UPS analysis are 2.2 eV and 1.4 eV, respectively. Based on the valence band maximum, and band gap, the conduction band minima are obtained (−0.96 eV and −0.16 eV for ZnO and CuO, respectively). On the basis of band edge energies and heterojunction construction revealed by HR-TEM, a mechanism is proposed in the Figure 11. 

Upon irradiation of the ZSC nanocomposite with simulated sunlight, electrons are excited from the VB of ZnO and CuO to their respective CB. Since the band edge potential of the CB of ZnO and CuO are −0.96 eV and −0.16 eV, respectively, and the work function of SWCNTs is −4.8 eV [9], the electrons are readily transferred from the CBs of ZnO and CuO to the SWCNTs, slowing the recombination of the charged species. The VB edge potential of pristine ZnO is 2.2 eV, which is higher than the oxidation potential of OH^−^ (to OH^•^) (1.99 eV). Thus, OH^−^ ions are oxidized to OH^•^ by holes at the VB of ZnO. However, oxidation may not be possible at the VB of CuO due to the low VB edge potential (1.4 eV). In this state, reduction of O_2_ molecules to O_2_^−•^ takes place at the SWCNT. Thus, SWCNTs act as an electron medium and promote the separation of electron-hole pairs, resulting ina synergistic effect. In addition, SWCNTs in the heterojunction increase active sites by introducing kinks on the surface, further increasing the photocatalytic activity [9]. Figure 11 shows the proposed type II mechanism of the ZSC3 heteronanocomposite.

The following mechanism is proposed for the photocatalytic degradation of MB:$$\mathrm{ZnO}\overset{h\upsilon }{\Rightarrow }\mathrm{ZnO}\;\left(e^{-}+h^{+}\right)$$
$$\mathrm{CuO}\overset{h\upsilon }{\Rightarrow }\mathrm{CuO}\;\left(e^{-}+h^{+}\right)$$
$$H_{2}O+h^{+}{\left(\mathrm{VB}\right)}_{\mathrm{ZnO},\;\mathrm{CuO}}⇌{\mathrm{OH}}^{•}+H^{+}$$
$$\left(\mathrm{SWCNT}\right){\;O}_{2}+e^{-}⇌O_{2}^{-•}$$
$$h^{+}\left(\mathrm{ZnO},\;\mathrm{CuO}\right)+{\mathrm{OH}}^{-}⇌{\mathrm{OH}}^{•}$$
$$O_{2}^{-•}+H^{+}⇌{\mathrm{HO}}_{2}^{•}$$
$${\mathrm{HO}}_{2}^{•}+{\mathrm{HO}}_{2}^{•}⇌H_{2}O_{2}+O_{2}$$
$$e^{-}+H_{2}O_{2}⇌{\mathrm{OH}}^{•}+{\mathrm{OH}}^{•}$$
$$H_{2}O_{2}+O_{2}^{-•}⇌{\mathrm{OH}}^{•}+{\mathrm{OH}}^{-}+O_{2}$$
$$H_{2}O_{2}\overset{h\upsilon }{\Rightarrow }2{\mathrm{OH}}^{•}$$
$$\mathrm{Dye}\;\left(\mathrm{MB},\mathrm{CR},\mathrm{RhB}\right)+{\mathrm{OH}}^{•}⇌\mathrm{Intermediates}⇌{\mathrm{CO}}_{2}+H_{2}O$$

## 3. Materials and Method

### 3.1. Chemicals

ZAD (99%, Sigma-Aldrich, St. Louis, Mo, USA), SWCNTs (>90%, US Research Nanomaterials, Houston, TX, USA) with a 1–2 nm outer diameter, CAH (98%, Sigma-Aldrich), and ethanol (200 proof, ACS reagent, ≥99.5%, Sigma-Aldrich) were used as supplied. Whatman 42 filter papers were used for the filtration processes.

### 3.2. Synthesis of ZnO–SWCNT–CuO Heteronanocomposites

A series of ZSC heteronanocomposites were prepared by reaction of ZAD and CAH, along with SWCNTs. A homogeneous solution was obtained by sonicating CAH and ZAD in 50 mL of ethanol for 1 h at 25 °C. CAH and ZAD were combined in ratios of 1:3, 1:6, 1:9, and 1:12 by mass for the synthesis of ZSC1, ZSC2, ZSC3, and ZSC4, respectively. To each of these homogenous solutions, 50 mg of SWCNTs was added. The resultant mixture was magnetically agitated at 25 °C for 1 h at a speed of 70 rpm. The obtained suspension was left for 8 h to ensure complete recrystallization and was then vacuum filtered. The filtered solid (photocatalyst) was placed in a preheated oven at 60 °C for 2 h to ensure complete evaporation of the ethyl alcohol. The dried photocatalyst was loaded into an alumina container with a cover, and the alumina container was loaded into a stainless-steel chamber (SUS316). The chamber was capped airtight using highly purified (oxygen-free and high-conductivity, OFHC) Cu gaskets. The locked chamber was inserted into a muffle-type furnace and annealed at 520 °C for 30 min to form the ZSC nanocomposites. The heating rate of the furnace was maintained at an optimum level (5 °C min^−1^). The details of the ZSC4 sample are not presented here because of its low photocatalytic efficacy.

### 3.3. Characterization

The crystallographic features of the nanocomposites were evaluated by XRD analysis (a scan rate of 5° min^−1^, range 10–80°, Cu Kα, λ = 0.154 nm, Smart Lab, Rigaku, Japan). The morphological features were studied by FE-SEM (SU-8230, Hitachi, Tokyo, Japan) and HR-TEM (JEM-2200FS, JEOL, Tokyo, Japan). The spectroscopic features were investigated by UV–Vis spectrophotometry (Lambda 25, PerkinElmer, Waltham, MA, USA) and high-performance XPS (Al Kα/1486.6 eV/Nexsa XPS system, Thermo Scientific, Waltham, MA, USA). The photoluminescence (PL) properties of the heteronanocomposites were studied by means of a fluorometer (λ_ex_ = 275 nm/LS55, PerkinElmer, Waltham, MA, USA). The chemical bonds and functional groups present in the nanocomposites were detailed using FTIR spectroscopy (Nicolet iS5, Thermo Scientific, Waltham, MA, USA). The LC-MS technique (6410 Triple Quad, Agilent Technologies, Santa Clara, CA, USA) was used to detect degradation fragments of the MB.

### 3.4. Preparation of Pure ZnO Wurtzite and Pure CuO Monoclinic Nanocomposites

Pristine ZnO nanorods (wurtzite) and pristine CuO monoclinic nanocrystals were prepared using ZAD and CAH by recrystallization followed by a sealed annealing method conducted via the same procedure used for the synthesis of ZSC nanocomposites. Pristine samples of ZnO and CuO were employed as controls during the assessment and study of the photocatalytic efficiency of the heteronanocomposites.

### 3.5. Thermal Stability Assessment

Thermogravimetric analysis (TGA) and differential scanning calorimetric analysis (DSC) of all the samples, including the SWCNTs, were performed using Q600 (Waters, New Castle, DE, USA) and Q20 (Waters, USA) thermal analyzers, respectively, to analyze the thermal stability of samples. All samples except the SWCNTs were investigated by heating at a ramp speed of 10 °C min^−1^ from 27° to 850 °C with the sample under an N_2_ atmosphere. For the SWCNTs, a O_2_ atmosphere was used.

### 3.6. Photocatalytic Performance of the Prepared ZnO–SWCNT–CuO Heteronanocomposites

The photocatalytic activities of the ZSC nanocomposites were investigated using two azo dyes (MB and CR) and one xanthene dye (RhB). Photocatalytic experiments were carried out under simulated sunlight generated by a Xe lamp (OSRAM XBO R, 300 W/60C, Germany). In each test experiment, 50.0 mg of a ZSC heteronanocomposite and 0.0025 g of dye (each) per 100 mL of solution were used. For the photocatalytic experiments, 50.0 mg of a ZSC nanocomposite was added to 100 mL (25 mg/L) of MB solution (pH = 6.63). No additional chemicals were used to adjust pH. The obtained photocatalyst/dye mixture was further sonicated for 0.5 h, followed by magnetic agitating for 45 min at 70 rpm. The suspension was then left in the dark for 1 h to achieve adsorption–desorption equilibrium among active sites of the heteronanocomposite and the dyes. For testing, the reaction system was placed at 15 cm from the light source. A quartz plate was used as a window to prevent evaporation. The light power per unit area was calculated to be approximately 0.11 W/cm^2^. The irradiation was conducted in the absence of external oxidizing agents (e.g., H_2_O_2_ or O_3_) at ambient temperature-pressure conditions. The reaction temperature was maintained below 35 °C using a cooling fan during the experiment. Aliquots were collected at intervals of 5 min to record the rate of photocatalytic reaction. The photodegradation progress was observed visually and was determined quantitatively by spectroscopic measurement. The absorbance was measured in the wavelength region from 200 to 800 nm. The experiments were performed for all the ZSC composites and for pristine CuO and pristine ZnO. The other experimental parameters were kept constant for all of the samples during the study.

### 3.7. Stability Assessment

The ZSC nanocomposites were collected after the completion of each cycle of photocatalytic degradation by vacuum filtration, washed several times with distilled water, dried at 100 °C for 24 h in an oven, and then used in a successive photocatalytic degradation experiment. The molar ratio of dye and ZSC was kept constant in each cycle. The process was continued for five cycles. The stability of ZSC was further investigated by XRD after its use.

### 3.8. Point of Zero Charge

The point of zero charge (pH_PZC_) was evaluated to investigate the charge adsorption on the nanocomposite surface as a function of pH. To determine the pH_PZC_, NaCl solutions of different pH within the range of 2–12 were prepared with a pH interval of 2. The solutions were prepared with 0.01 N NaCl and 0.1 N HCl/NaOH solutions. Ten milligrams of ZSC were added to 20 mL of pH-adjusted NaCl solution each time, followed by gentle stirring for 24 h before recording the final pH. 

### 3.9. Charge-Carrier Trapping Experiments

Scavenging agents were used in the photocatalytic examination of ZSC to trap the photoactive OH^•^ radicals, holes, and superoxide radicals. The scavengers used were isopropyl alcohol (3 mL), potassium iodide (5 mg), and 1,4-benzoquinone (5 mg) for hydroxyl radicals, holes, and superoxide radicals, respectively.

### 3.10. Electrochemical Analysis

Electrochemical measurements were carried out using an electrochemical workstation (ZIVE SP1, WonATech, Seoul, Korea). Standard three-electrode cell (working electrode: sample, counter electrode: platinum. In addition, reference electrode: saturated Ag/AgCl) was used. To make a working electrode, a colloidal solution of the sample in an ethanolic solution of Nafion was taken, which is coated on a glassy carbon electrode (d = 3 mm, MF-2012, BASI, West Lafayette, IN, USA). Aqueous sodium sulfate (0.2 M) solution was used as an electrolyte.

## 4. Conclusions

As photodegradation catalysts for various dyes, a series of ternary ZSC nanocomposites were prepared via a two-step synthesis method of recrystallization followed by annealing in a sealed vessel. The formation of heterojunctions of three crystal structures adjacent to each other was revealed. In the heterojunctions of ZnO–SWCNT–CuO, ZnO and CuO were linked together with SWCNTs. The ZSC composites showed nearly 5 times improved photocatalytic activity compared to ZnO due to the increase in light absorption, electron-hole separation, oxygen vacancies, and other defects on the composite surface. The best composite ZSC3 has excellent degradation stability up to 5 cycles. In addition, it was determined that OH^•^ has a decisive effect on dye degradation, while O_2_^−•^ and h^+^ were involved to a lesser extent. A type II heterojunction mechanism pattern is proposed based on the charge carrier migration pattern. The CV measurement showed that the heterojunction construction retained a higher oxidation potential for two main crystal structures. This indicates the potential of the ZSC samples in other applications where higher reduction/oxidation potential is required, such as water splitting. The ZSC3 shows the best photocatalytic performance under simulated sunlight with high stability.

## Figures and Tables

**Figure 1 molecules-27-08812-f001:**
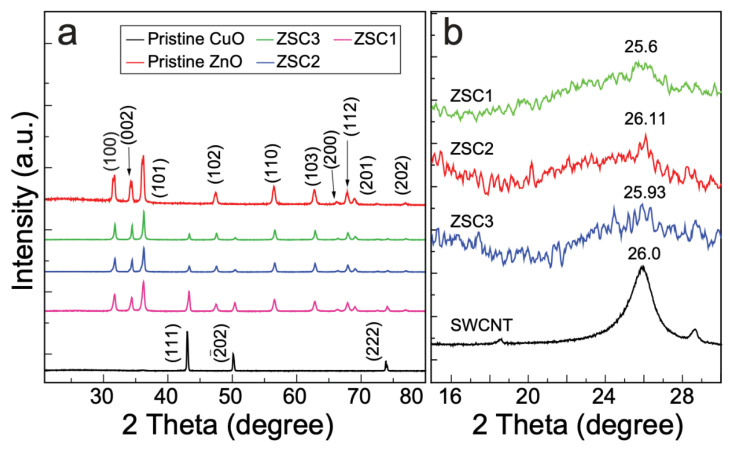
(**a**) XRD plots of the pristine oxides and heteronanocomposites. (**b**) The magnified XRD patterns showing a peak of crystalline carbon, confirming the presence of SWCNTs in the heterojunction nanocomposites.

**Figure 2 molecules-27-08812-f002:**
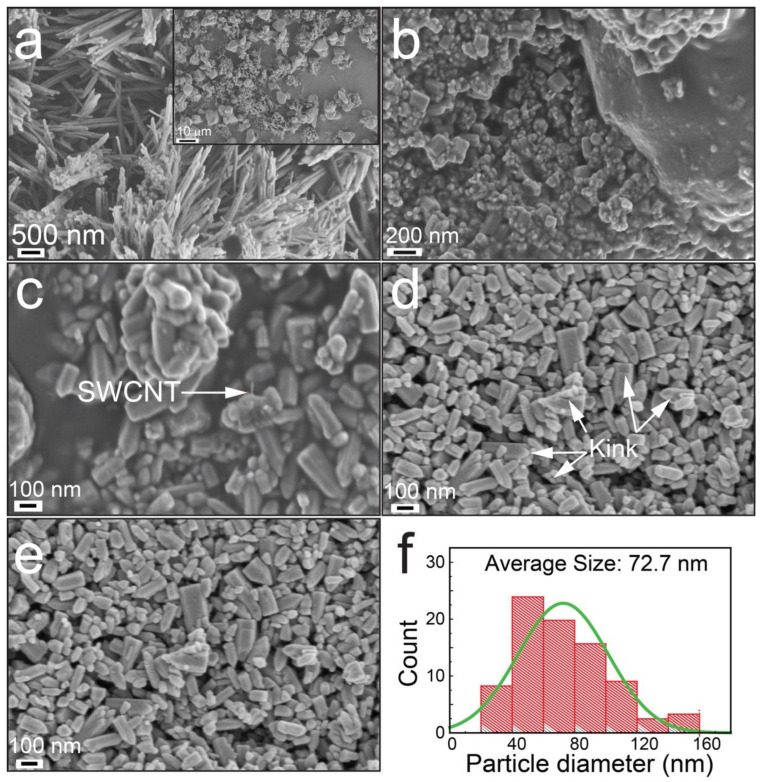
FE-SEM images: (**a**) pristine ZnO, (**b**) ZSC1, and (**c**) ZSC2. (**d**) Micrograph showing defects developed on the surface of the ZSC3 heteronanocomposite. (**e**) Micrograph of ZSC3. (**f**) Histogram of the particle size distribution of the ZSC3 heteronanocomposite. The inset in (**a**) shows the FE-SEM image of pristine monoclinic CuO nanocrystals.

**Figure 3 molecules-27-08812-f003:**
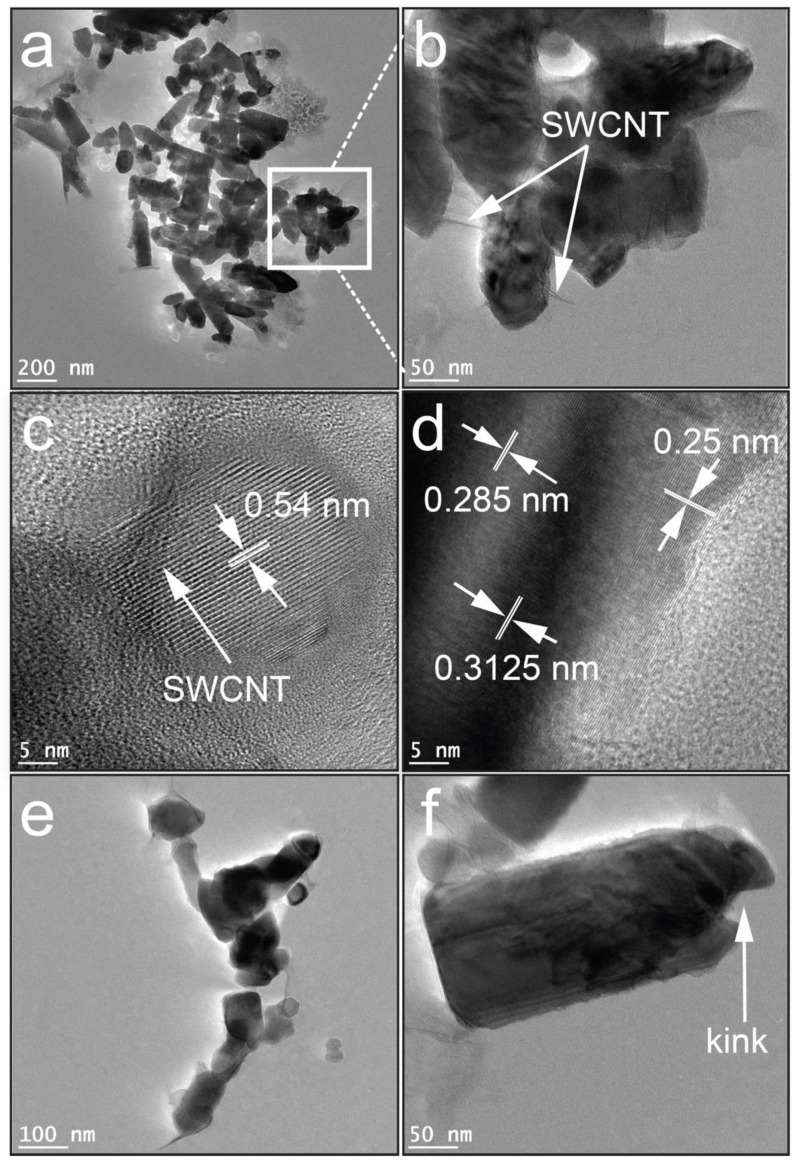
HR-TEM images of the ZSC nanocomposites: (**a**) ZSC2, (**b**) enlarged micrograph of Figure 3a, (**c**,**d**) lattice fringes of the ZSC3 nanocomposite, (**e**) magnified view of the ZSC3 nanoparticles, and (**f**) enlarged micrograph of a single ZSC3 nanoparticle with a defect.

**Figure 4 molecules-27-08812-f004:**
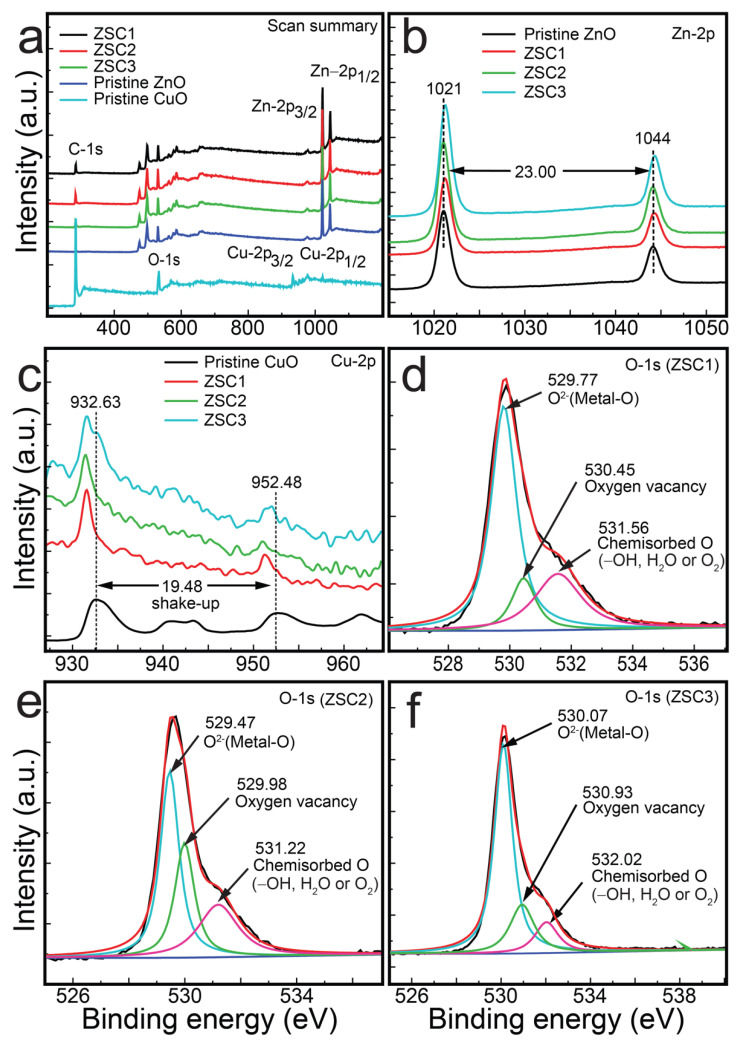
XPS spectra of ZSC nanocomposites and pristine samples: (**a**) survey spectra, (**b**) Zn-2p_3/2_ and Zn-2p_1/2_ spectra of ZSC nanocomposites and pristine ZnO, (**c**) Cu-2p_3/2_ and Cu-2p_1/2_ spectra of ZSC nanocomposites and pristine CuO, (**d**) O-1s spectrum of ZSC1, and (**e**) O-1s spectrum of ZSC2, (**f**) O-1s spectrum of ZSC3.

**Figure 5 molecules-27-08812-f005:**
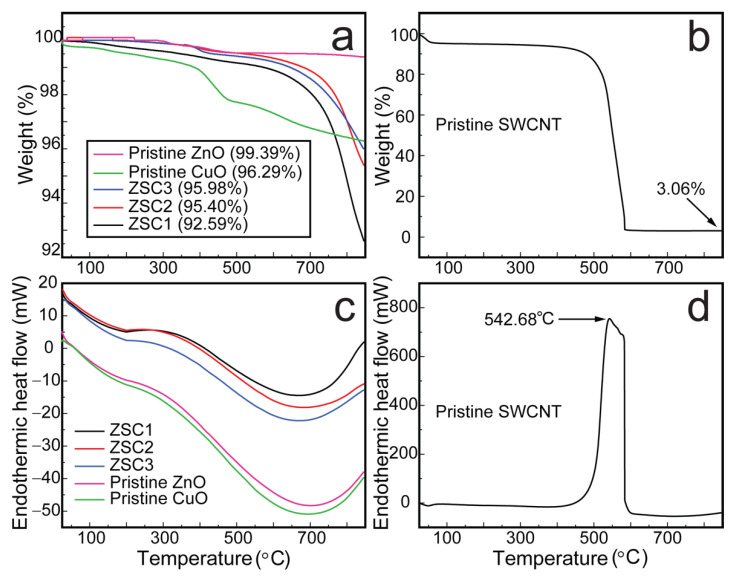
The thermal analysis of ZSC heteronanocomposites and pristine samples. The thermogravimetry plots of (**a**) the heteronanocomposites, pristine ZnO, and pristine CuO and (**b**) pristine SWCNTs. The differential scanning calorimetry thermograms of (**c**) the heteronanocomposites, pristine ZnO, and pristine CuO, and (**d**) pristine SWCNTs.

**Figure 6 molecules-27-08812-f006:**
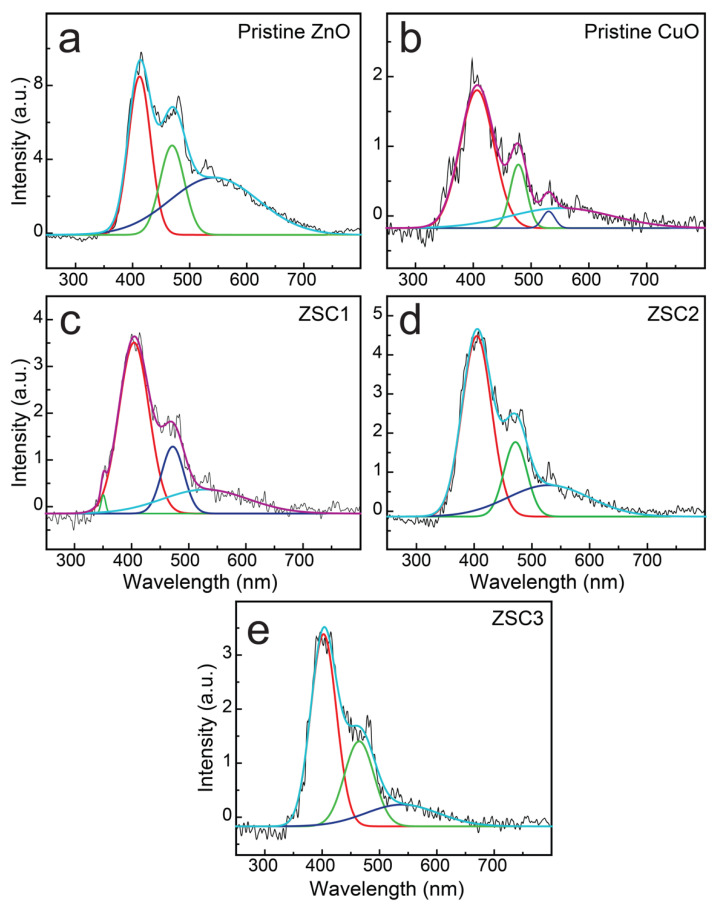
PL spectra for (**a**) pristine ZnO, (**b**) pristine CuO, (**c**) ZSC1, (**d**) ZSC2, and (**e**) ZSC3. Each spectrum is deconvoluted and the deconvoluted spectra are displayed together with each original spectrum.

**Figure 7 molecules-27-08812-f007:**
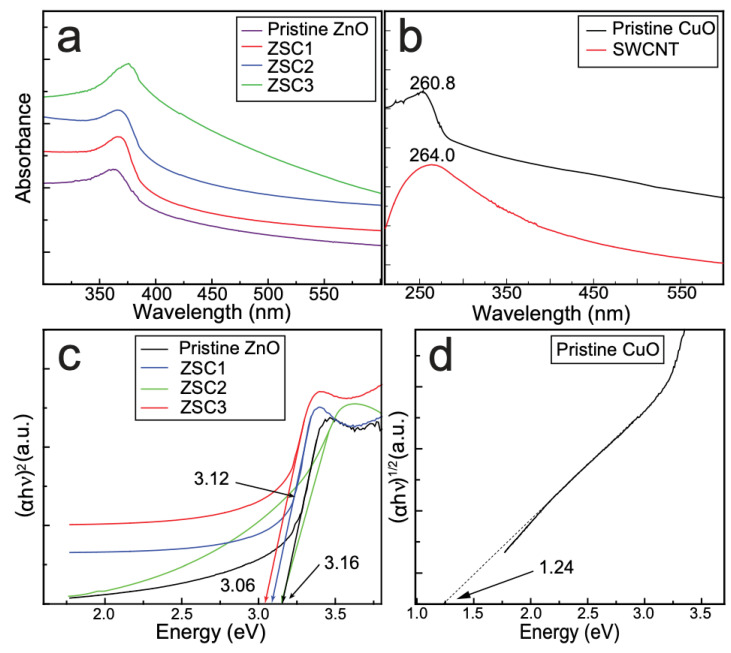
(**a**) The combined plot of the UV–Vis spectra of pristine ZnO, ZSC1, ZSC2, and ZSC3. (**b**) The absorption spectra of pristine CuO, and SWCNTs. (**c**) Tauc plots obtained by applying the direct-bandgap method, depicting the bandgap energies of the pristine ZnO and ZSC composites. (**d**) Tauc plot for pristine CuO obtained by applying the indirect bandgap method.

**Figure 8 molecules-27-08812-f008:**
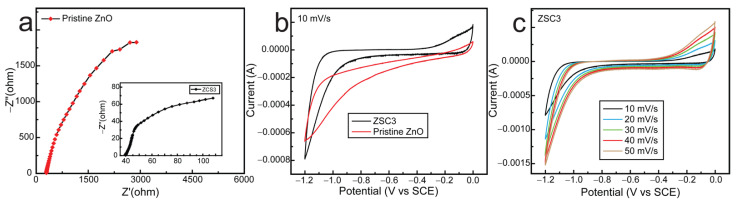
(**a**) EIS of pristine ZnO and ZSC3, (**b**) CV curves for ZSC3 and pristine ZnO, (**c**) CV curves of ZSC3 at different scan rates.

**Figure 9 molecules-27-08812-f009:**
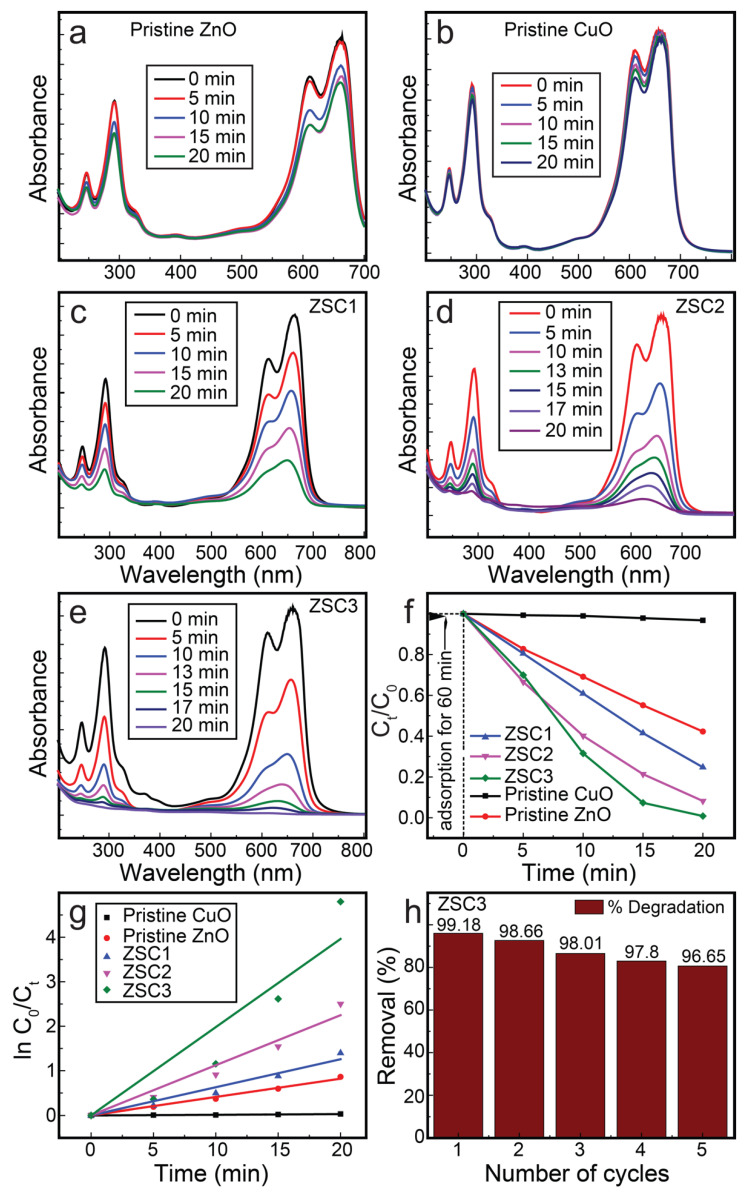
The spectrometric scans for the photocatalytic decomposition of MB in the presence of (**a**) pristine wurtzite ZnO, (**b**) pristine monoclinic CuO, (**c**) ZSC1, (**d**) ZSC2, and (**e**) ZSC3. (**f**) Concentration degradation plot as a function of time. (**g**) Kinetic plot for all the samples. (**h**) The stability results for ZSC3 heteronanocomposite.

**Figure 10 molecules-27-08812-f010:**
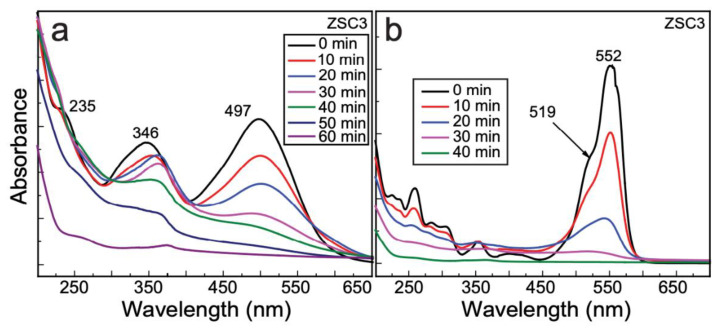
The photocatalytic degradation of dyes (**a**) Congo red and (**b**) rhodamine B by the ZSC3 nanocomposite.

**Figure 11 molecules-27-08812-f011:**
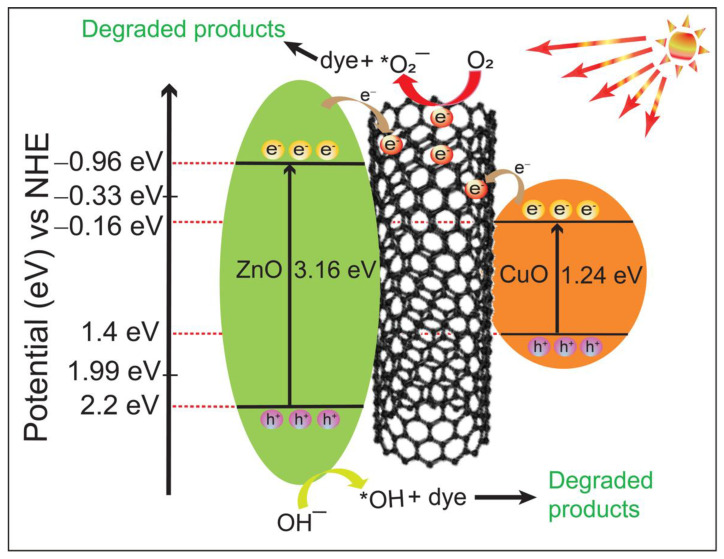
The bandgap structure and scheme for photoexcitation of e–h pairs, revealing the catalytic action of the ZSC3 heteronanocomposite.

**Table 1 molecules-27-08812-t001:** The rate constant (*k*, min^−1^) for photodegradation of dyes (MB, CR, RhB).

Sample	Dye	Concentration (ppm)	Rate Constant, K (min^−1^)	Increased Rate (Multiple)
Blank	MB	25	0.0027	1
Pure ZnO	MB	25	0.0410	15.21
ZSC1	MB	25	0.0629	23.31
ZSC2	MB	25	0.1124	41.63
ZSC3	MB	25	0.1981	73.35
ZSC3	CR	25	0.046	NC *
ZSC3	RhB	25	0.137	NC *
Pure TiO_2_	MB	25	0.099	36.66

* NC = not compared.

**Table 2 molecules-27-08812-t002:** The comparison of the photocatalytic performance of our catalyst with those of catalysts reported in the literature.

Catalyst	Pollutant (Dye)	Dose of Dye	Catalyst Dose (g)	Light Source	Degradation (%)	Time (min)	Ref.
ZSC3	MB	25 ppm	0.05	Simulated sunlight	99.18	20	Present work
	CR		0.05		94.13	60	
	RhB		0.05		99.63	40	
	MB		0.05	Visible	76.7	70	
CuO/ZnO	MB	0.025 g/L	0.02		98.6	80	[6]
ZnO/MWCNTs	MB	10 mg/L	0.01	UV	NA	30	[16]
α-Fe_2_O_3_/graphene	MB	40 mg/L	0.1	UV	NA	80	[17]
CuO/SWCNT	MB	25 mg/L	0.1	Sunlight	96	120	[13]
CuO/ZnO/eggshell membrane	CR	NA	0.004	UV–Vis	NA	120	[14]
CuO doped ZnO-CNTs	RhB	2 mg/L	200	UV	NA	70	[15]
Pure TiO2	MB	10 mg/L	0.1	Xenon	51	180	[18]
ZnO-CuO/ES	CR	10 mg/L	0.1	Visible	83	240	[19]
Ag@ZnO	MB	10 mg/L	NA	Simulated sunlight	99.3	90	[20]
	CR	20 mg/L			98.5	55	
NaBH_4_-ZnS	RhB	4 mg/L	0.02	UV	93.6	210	[21]
Cu_2_O	RhB	5 ppm	0.1	UV	97	200	[22]

**Table 3 molecules-27-08812-t003:** The inhibition characteristics of different scavengers’ kinetics.

Scavengers	Reactive Species Quenched	*K* (min^−1^)	Inhibition Rate (%)
Control	-	0.198	0
IPA	OH^•^	0.011	95
KI	*h* ^+^	0.032	86
BQ	O_2_^•^	0.168	29

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
