# Peer review of "Carbon-Based Ternary Nanocomposite: Bullet Type ZnO–SWCNT–CuO for Substantial Solar-Driven Photocatalytic Decomposition of Aqueous Organic Contaminants"

_molecules, 2022, doi:10.3390/molecules27248812_

Round 1

Reviewer 1 Report

This manuscript reported the synthesis, characterization and photocatalytic performance of ZnO–SWCNT–CuO heterojunction towards dye discoloration. Although study shows promising results, there are some issues that have to be improved before considered for publication. The detailed comments are shown below.

1.      As there is no method confirming degradation intermediate like TOC, I highly recommend authors use “discoloration” instead “degradation” or “decomposition” throughout manuscript.

2.      p.2, r.87-92: It would be nice to involve contribution of CNT in photocatalytic process more specifically.

3.      p.5, r. 210 correct (-202)

4.      In Fig. 1b, XRD peak related to CNT is disputative. Show XRD patterns of pristine CNT for comparison of its position.

5.      Fig. 2. Adjust contrast and brightness of micrographs to same level.

6.      Is there any reason to discuss broadly thermal stability/properties? Results shown in Fig. 5a seems to be inconsistent.

6.      Data presented in Fig. 6 should be improved. Peaks fitting is good practice.

7.      I highly recommend authors to rearrange Fig.7 by using data given in S4 and S5. It would be more clear to reader if you show all relevant data for both components and composites.

8.      The numbers should be better rounded. Page p.15, r.471. but also p.6, r. 260.

9.      Fig.9. The rate constant increase for all heterostructure ZC1-3 in comparison to single component is outstanding and seems to be more than synergic effect. It is highly desirable to provide dye adsorption and photolysis in the presence of CNT. Keep in mind that adsorption capacity of CNT may alter significantly under irradiation.  

10.      Concerning reusability study; it is good practice to show at least XRD patterns of heterostructure after several consecutive photocatalytic runs (see for example 10.1016/j.matchemphys.2019.121823). Although the reusability is satisfactory, it can be accompanied with photocatalysts phase structure change. Verify.

11.      Concept of solid state heterostructure given in Fig. 11 needs revision. The band edge positions of CuO to relative to ZnO is wrong. If electrons from ZnO CB falls to CNT, how can they reduced oxygen? What is the role of holes in CuO VB?  

All these points can really improve the scientific level of the results and the article. So a major revision is needed for it.

Author Response

The response is attached.

Reviewer 2 Report

The manuscript (Ref. molecules-2021154) describes the synthesis of different hybrid photocatalysts based on CuO and ZnO doped with carbon nanotubes, their characterization, and their subsequent use in the photoremoval of different dyes. I would firstly like to commend the authors on their delivering a manuscript that reads very well thanks to a highly polished English usage. There are some minor mistakes (see below) that may be corrected easily but otherwise the manuscript makes for a very easy read. ON the other hand, its scientific robustness is somewhat weaker due to a series of fundamental issues that severely hinder the overall impact of the manuscript. In its present state, I am sorry I am unable to recommend the paper for publication. The authors are advised to consider the following points to come up with a more solid and thorough manuscript that would result in a deeper insight into their very interesting experimental results.

1) Concerning English usage, the authors should correct the following mistakes:

- Line 55: “necessitates” should read “requires”

- Line 56 “releasing” should read “its release”

- Line 69 “its activity in visible” should read “its activity under visible”

- Line 72-73 “enables to control of” should read “enables control of”

- Line 79 “improved catalytic activity than the individual” should read “improved catalytic activity over the individual”

- Please avoid using the saxon genitive (e.g. line 334 “on the sample’s surface”)

2) Lines 126-128 “The filtrate was placed in a preheated oven at 60°C for 2 h to ensure complete evaporation of the ethyl alcohol.” I fail to understand this. The filtrate is by definition "a liquid which has passed through a filter", but this would make no sense. If I understand it correctly, the authors dry the filtered solid (not the filtrate) in an oven at 60ºC and then anneal the resulting solid at 520ºC (naming it dried residue is very confusing, how can your desired product be called a "residue"?). Could the authors clarify this point?

3) Line 135 Please provide more experimental details for characterization experiments, such as irradiation source or excitation wavelength, step time and rate, etc.

4) Line 159, when preparing the MB solutions was the pH adjusted by adding some chemical? This should be specified.

5) Line 170. “The intensity of light was approximately 0.11 W/cm-2.” How was this measured?

6) Lines 225-226. The average crystallite size was measured for which phase? There are two oxides present in the sample.

7) Section 3.2 It is impossible to detect wurtzite or monoclinic structures using SEM unless HR-SEM is used. At best you may mention their needle-like shape in the case of ZnO, but their crystal structure cannot be established by conventional electron microsocpy. The same applies to the presence of kinks in the sample, which are discernible through TEM, not SEM.

8) Lines 259-260. The difference in crystal sizes with the value obtained by XRD is quite high. Could the authors comment on this?

9) In Figure 2b, arrows pointing at SWCNTs are not necessary.

10) Line 281, Figures 3e and 3f show the presence of SWCNTs with a hexagonal wurtzite structure. Electron diffraction data (not provided) is needed to make such a claim.

11) Lines 370-374. How can the authors distinguish between broad bands and sharp peaks using spectra with such poor spectral resolution? In order to make such claims a sharper spectrum is needed and a suitable peak fitting is mandatory.

12) Figure 8a. Would it be possible to plot the two graphs together for a better visualization?

13) Line 421. “Among the ZSC nanocomposites, ZSC3 exhibited the largest bandgap lowering to 3.06 eV.” Do the authors mean “smallest”?

14) Line 545 “These results agree with the experimental pHPZC data” How do the authors reach such a conclusion? Could they ellaborate their reasoning? I fail to see a connection between pHpzc and the influence of the radical species.

15) The authors should use some experimental technique such as UPS or high resolution XPS to verify the position of the highest level of the conduction band.

16) Line 576 “electron transfer occurs from the CB of the CuO to the VB of the ZnO through the SWCNTs via an indirect dual Z-scheme.” If this is the case, OH· radical formation would be impeded according to Figure 11, which is the opposite of what the authors observe. In order to make these claims the band structure of the SWCNTs must be presented. Figure 11 is incomplete without this information.

17) This proposed mechanism (lines 586-596) needs to specify the location of both electrons and holes throughout the process

Author Response

The response is attached.

Reviewer 3 Report

Manuscript Number: molecules-2021154

Title: “Carbon-based dual Z-Scheme ternary nanocomposite: bullet type ZnO–SWCNT–CuO for substantial solar-driven photo-catalytic decomposition of aqueous organic contaminants”

Authors: Santu Shrestha, Kamal Prasad Sapkota, Insup Lee, Akherul Islam, Anil Pandey, Narayan Gyawali, Jeasmin Akter, Harshavardhan Mohan, Taeho Shin, Sukmin Jeong and Jae Ryang Hahn

The manuscript studies the photodecomposition of three different dyes under simulated sunlight: Methylene blue (MB), Congo red (CR) and Rhodamine B (RhB). To carry out the photocatalytic reactions, the authors synthesized and characterized ternary hetero-composites of ZnO, CuO and carbon nanotubes. They showed that, of all the synthesized compounds, the sample with hydrated copper(II) acetate and dihydrated zinc(II) acetate in a predetermined proportion (ZSC3), was the one that exhibited the highest photocatalytic activity. Finally, they demonstrated that these nanocomposites presented very good reusability and stability, with high levels of activity after five successive tests. I suggest publication of the manuscript after the authors consider the following comments and recommendations:

Lines 159-165, Section 2.6:

To make clear to readers the experimental device utilized, it would be convenient to give more details about the solar simulator-photocatalytic reactor system. For example, was any device used to keep the working temperature constant? Was a window used on the reactor to prevent losses of reactants? If so, what material and optical properties were used for this reactor window? Probably, a simple scheme or representation of the experimental setup can be helpful.

Lines 169-170:

In this paragraph, the authors declared "For testing, the reaction system was placed at 15 cm from the light 169 source. The intensity of light was approximately 0.11 W/cm-2". They have to report the main characteristics of the instrument used to measure the intensity of light at the output of the homemade simulated sunlight device.

Line 170 and line 439:

Note that “W/cm-2” should be replaced by "W/cm2".

Lines 403-404 and Figure S4(a):

In Figure S4(a), the authors showed a combined plot of the UV–Vis spectra of ZSC1, ZSC2, and ZSC3 and the absorption spectrum for SWCNTs (inset). A good absorbance is observed for low wavelengths at 250 nm and an absorption peak at 264 nm. For these low wavelength values, Direct Photolysis of the contaminant can occur; that is, the pollutant degradation without the presence of a photocatalyst. Have they verified the existence (or absence) of direct photolysis reaction in the experimental runs?

Line 432, Section 3.5. Photocatalytic activity:

In this section, the authors should explain whether the possible leaching of Zn or Cu from the synthesized photocatalyst was studied and the methodology used to perform the verification.

Lines 468-469 (Figure S7) and lines 524-515 (Figure S8):

In the linear regression of Figures S7 and S8, the value of R-Squared (R² or the coefficient of determination) should be reported, especially in Figure S8 for the dye Rhodamine B (RhB).

Lines 490-492 (Figure 10a) and lines 506-508 (Figure 10b):

In Figure 10a and b, the authors show the photocatalytic degradation of two dyes, Congo red and Rhodamine B, produced by the ZSC3 nanocomposite. They found that by increasing the irradiation time (60 min for CR and 40 min for RhB) complete decolourization of the dye occurs. Note that, although complete decolourization of the dye occurs at high wavelengths, at low wavelengths (approximately between 200 and 300 nm) they have radiation absorption and this effect can generate intermediate compounds that could be more toxic than the original pollutant. The authors must clarify this point.

Lines 549-550 (Figure S10) and Tables 1 and 3:

In the linear regression of Figure S10, the value of R-Squared must be reported for the synthesized nanocomposite designated as ZSC3. It must be noted that a good linear correlation was not obtained for ZSC3. Furthermore, the calculated values of rate constants (k) for MB degradation by ZSC1, ZSC2, and ZSC3 are reported in Table 1, and the same linear regression value k=0.1981 1/min is used to select the most efficient synthesized ZSC. Subsequently, Table 3 shows the constant rate and inhibition rate (%) of MB photodegradation reaction mediated by different scavengers, and the value 0.198 1/min is used again as a control to compare the inhibition rates.

The authors must clarify this important point.

Lines 567-568:

In this paragraph, the authors have cited Fig. 10. Is the number of the cited figure correct?

Author Response

The response is attached.

Round 2

Reviewer 1 Report

All issues raised in the review were responded. I propose to accept revised manuscript in present form.

Reviewer 2 Report

The authors have undertaken significant efforts in producing the revised version of the manuscript, which is now acceptable for publication.